# A Novel Urine Exosomal lncRNA Assay to Improve the Detection of Prostate Cancer at Initial Biopsy: A Retrospective Multicenter Diagnostic Feasibility Study

**DOI:** 10.3390/cancers13164075

**Published:** 2021-08-13

**Authors:** Yun Li, Jin Ji, Ji Lyu, Xin Jin, Xing He, Shaojia Mo, Huan Xu, Jingyi He, Zhi Cao, Xi Chen, Yalong Xu, Lei Wang, Fubo Wang

**Affiliations:** 1Department of Urology, Shanghai Shibei Hospital of Jingan District, Shanghai 200435, China; liyun@smmu.edu.cn; 2Department of Urology, Changhai Hospital, Second Military Medical University, Shanghai 200433, China; jijin@smmu.edu.cn (J.J.); lvji@med.uestc.edu.cn (J.L.); hexingqc@smmu.edu.cn (X.H.); 120094@sh9hospital.org.cn (H.X.); caozhi@smmu.edu.cn (Z.C.); 12340700016@fudan.edu.cn (X.C.); 12340700012@fudan.edu.cn (Y.X.); 3Clinical Immunology Translational Medicine Key Laboratory of Sichuan Province, Organ Transplant Research Institute, Sichuan Provincial People’s Hospital, University of Electronic Science and Technology of China, Chengdu 610072, China; 4Department of Urology, Zhongda Hospital, Southeast University, Nanjing 210044, China; 230199231@seu.edu.cn; 5Department of Urology, Taizhou People’s Hospital, Taizhou 225399, China; 6School of Basic Medicine, Second Military Medical University, Shanghai 200433, China; moshaojia@smmu.edu.cn; 7Department of Urology, Shanghai Ninth People’s Hospital, Shanghai 200011, China; 8Department of Urology, The First Affiliated Hospital of Soochow University, Suzhou 215006, China; 20184232040@stu.suda.edu.cn

**Keywords:** exosomes, prostate cancer, lncRNA assay, PCA3, MALAT1, diagnosis

## Abstract

**Simple Summary:**

Prostate cancer (PCa) is the second most common malignancy in males globally. Although PSA screening is a milestone in PCa detection, it also causes overdiagnosis and subsequent overtreatment. Therefore, it is imperative to find an optimal replacement or supplement for PSA testing to increase the detection rate of clinically significant PCa as well as reduce unnecessary biopsies. Here, we aimed at developing and validating a novel noninvasive urinary exosome-based post-DRE lncRNA assay to diagnose PCa and clinically significant PCa at initial prostate biopsy. We found that the lncRNA assay had a significant clinical value in diagnosing PCa and clinically significant PCa compared to the current clinical parameters. These results suggest that this novel lncRNA assay developed in this study could be a valuable biomarker to increase the detection rate of clinically significant PCa as well as reduce unnecessary biopsies.

**Abstract:**

Purpose: This study aimed at developing and validating a novel noninvasive urinary exosome-based post-DRE (digital rectal examination) lncRNA assay to diagnose PCa (prostate cancer) and clinically significant PCa (Gleason score ≥ 7) from the initial prostate biopsy. Methods: A total of 602 urine samples from eligible participants were collected. The expression levels of urinary exosomal PCA3 (prostate cancer antigen 3) and MALAT1 (metastasis-associated lung adenocarcinoma transcript 1) were detected by qPCR (quantitative real-time PCR). Receiver operating characteristic (ROC) analysis was applied to evaluate the diagnostic performance of PCA3, MALAT1 and the lncRNA assay. A decision curve analysis (DCA) and waterfall plots were used to assess the clinical value of the lncRNA assay. Results: Urinary exosomal PCA3 and MALAT1 were overexpressed in PCa and clinically significant PCa (*p* < 0.001). The lncRNA assay combining PCA3 and MALAT1 had a better diagnostic performance (AUC 0.828) than the current clinical parameters in detecting PCa. More importantly, the lncRNA assay yielded an AUC of 0.831 to detect clinically significant PCa, which is much higher than that of the current clinical parameters. The lncRNA assay was superior to PSA, f/tPSA and the base model for detecting PCa and clinically significant PCa, with a higher net benefit for almost all threshold probabilities. At the cutoff value of 95% sensitivity, the lncRNA assay could avoid 24.2% unnecessary biopsies while only missing 1.2% of the cases of clinically significant PCa. Conclusion: We developed and validated a novel noninvasive post-DRE urine-based lncRNA assay that presented good diagnostic power and clinical utility for the early diagnosis of PCa and high-grade PCa.

## 1. Introduction

Prostate cancer (PCa) is one of the leading causes of death in men worldwide. It was estimated that there would be 1,414,259 new cases and 375,304 new deaths in 2020 globally [1]. Since the introduction of prostate-specific antigen (PSA) testing, PCa incidence has increased dramatically, and appropriate treatment has significantly decreased PCa mortality. However, PSA’s routine use has been questioned recently due to its limited specificity despite good sensitivity, resulting in at least 60% unnecessary biopsies [2]. Moreover, PSA-initiated prostate biopsies have revealed excessive indolent PCa that may not require lifelong treatment, leading to overdiagnosis and subsequent overtreatment [3,4]. It is generally well-accepted that clinically significant PCa or high-grade PCa (GS (Gleason score) ≥ 7) benefit the most from treatment, which includes either radiotherapy or surgery. The diagnostic value of the PSA in clinically significant PCa is also limited. For these reasons, the United States Preventive Service Task Force (USPSTF) recommended against PSA screening in 2012. However, as there is currently no replacement for PSA for diagnosing clinically significant PCa, the USPSTF recommended age-specific shared-decision PSA testing in men aged 55–69 years [5]. Therefore, it is imperative to find an optimal replacement or supplement for PSA testing to increase the detection rate of clinically significant PCa and reduce unnecessary biopsies.

Urine, which is easily and noninvasively obtained, has been extensively explored as a source of diagnostic biomarkers. The Progensa PCA3 (prostate cancer antigen 3) assay was approved for men undergoing repeated biopsies. However, its efficacy over the PSA test is still limited (AUC (area under the curve) of 0.64 to 0.76, with AUC increments over PSA of 0–0.16) [6]. Other urinary biomarkers or panels have recently emerged, including the Michigan Prostate Score (MiPS) and SelectMDx [7]. Although these biomarkers have been well-validated, their association with clinical parameters needs to be further studied to enhance their diagnostic ability, and the clinical utility of these biomarkers still has a long way to go. Exosomes are small extracellular vesicles of 40~160 nm in diameter secreted by cells that carry various molecules and provide a promising noninvasive method for detecting cancers [8,9]. Recent studies have shown that circulating exosomal RNAs (exRNAs) could serve as promising biomarkers for cancer detection [9,10,11]. However, as an easily collected and noninvasive source of cancer biomarkers, urinary exosomes have not been adequately explored [12]. We previously evaluated the diagnostic performance of urinary MALAT1 [13] and PCA3 [14] in the Chinese population and demonstrated that both urinary MALAT1 and PCA3 could be useful biomarkers for the early detection of PCa. Due to the ease of obtaining and them being a noninvasive source of specimens and a stable detectable source of biomarkers, we proposed that urinary exosomal MALAT1 and PCA3 could be promising biomarkers for the improvement of PCa detection. Therefore, we first aimed at developing a standardized method for post-DRE urine collection, sample processing and RNA quantification. We then evaluated the expression levels of lncRNA PCA3 and MALAT1 (metastasis-associated lung adenocarcinoma transcript 1) in a large cohort of PCa and BPH (benign prostatic hyperplasia) participants. Finally, this study was designed to establish an lncRNA assay to detect PCa and clinically significant PCa during the initial prostate biopsy.

## 2. Methods

### 2.1. Sample Collection

The study was approved by the Ethics Committee of the Shanghai Changhai Hospital, the Shanghai Shibei Hospital and the Taizhou People’s Hospital (No. CHEC2013-115). All the sites shared the same standard operating procedure (SOP) for participant recruitment, sample processing and prostate biopsy. Written informed consent was obtained from the participants before sampling. First-catch urine samples had been collected following an attentive DRE (digital rectal examination) (three strokes per lobe) before the biopsy was performed. The urine samples were immediately cooled on ice and processed within two hours of collection [15]. The protocol of prostate biopsy was described previously [13]. The patients who underwent biopsies were included if they either had an increase in PSA greater than 4 ng/mL or PSA that was not elevated but who had a DRE that revealed a nodule or imaging examination abnormality. PCa and BPH samples were confirmed by prostate biopsies, and the pathology of the biopsy tissues was examined by two pathologists to confirm the diagnosis and the Gleason score.

### 2.2. Exosome Isolation

According to the manufacturer’s instructions, exosomes were isolated using a Wayen Exosome Isolation Kit (EIQ-03001(Urine), Wayen, Shanghai, China). In brief, the urine samples were kept on ice and then centrifuged at 3000× *g* for 15 min at 4 °C. Twenty milliliters of the urine supernatant were mixed with 7.5 mL of reagent A and 670 μL of reagent B. The mixture was incubated at 4 °C overnight (16 h) and then centrifuged at 3000× *g* for 1 h at 4 °C. One milliliter of the supernatant was transferred to a 1.5 mL EP tube, and the residual supernatant was discarded. The pellet was resuspended entirely with the 1 mL supernatant above. The resuspension was centrifuged at 10,000× *g* for 10 min at 4 °C, and the supernatant was discarded. The pellet was then resuspended with 20 μL of PBS. The sample was finally centrifuged at 10,000× *g* for 5 min at 4 °C. The supernatant, which contained exosomes, was collected and transferred to a 1.5 mL EP tube. The exosomes could be used for further research or stored at −80 °C.

### 2.3. Western Blots (WB)

A total of 20 µL of the RIPA lysis and extraction buffer (89901, Thermo Fisher, Waltham, MA, USA) and 0.2 µL of a protease inhibitor (B14001, Bimake, Houston, TX, USA) were added to the pellet. Then, the mixture was centrifuged at 12,000× *g* for 15 min at 4 °C. The supernatant was collected, and the protein concentration was measured using a BCA kit (23228, Thermo Fisher, Waltham, MA, USA). Next, a 5× loading buffer (C508320-001, Sangon, Shanghai, China) was added to the supernatant, and the mixture was heated in a metal bath (Thermomixer comfort, Eppendorf, Hamburg, Germany) at 100 °C for 15 min. According to the manufacturer’s instructions, running gels were prepared using a PAGE Gel Fast Preparation Kit (PG112, EpiZyme, Shanghai, China). Then, 20–40 μg exosome samples and a protein marker (1610374, Bio-Rad, Hercules, CA, USA) were added. The instrument was set to 100 V; the electrophoresis time was 1 h 45 min. After the electrophoresis, the gel was removed, attached to a PVDF (Millipore, Burlington, VT, USA) membrane and placed in a transfer tank. The instrument was set to 100 V; the transfer time was 1 h 30 min. After the transfer was completed, 5% BAS was used for blocking for 2 h at room temperature. After the membrane was cut at the appropriate position, the primary antibodies were added. These antibodies included CD9 (AP1482, Abgent, Suzhou, China), CD63 (AP5333, Abgent, Suzhou, China), CD81 (AM8567, Abgent, Suzhou, China), TSG101 (AM8662, Abgent, Suzhou, China) and ACTB (A5441, Sigma, St. Louis, MO, USA). The antibody was incubated with the membrane overnight at 4 °C. The next day, the membrane was removed and left at room temperature for 15 min. The primary antibody was removed and washed three times with 1× TBST for 5 min each time. Then, the secondary antibody was added, incubated for 2 h at room temperature and washed three times with TBST for 5 min each time. Finally, the pictures of blots were taken by the AI600 system (GE, Boston, MA, USA).

### 2.4. Nanoparticle Tracking Analysis (NTA)

The extracted exosomes from 10 mL of urine were resuspended in 150 μL of PBS (SH30256, HyClone, Waltham, MA, USA) (filtered). Before testing, the exosomes were diluted to 1 mL. The manufacturer’s instructions were followed to turn on Nano Sight 300 (Malvern, Malvern, UK). The detection module was cleaned automatically. Then, a syringe was used to aspirate air and purge three times. Then, the software was used routinely to detect the exosomes, and the results were saved.

### 2.5. Transmission Electron Microscopy (TEM)

The extracted exosomes from 10 mL of urine were resuspended in 200 μL. After mixing the sample, a pipette was used to aspirate 20 μL of the resuspension and place it onto a copper mesh. The drop was stopped when the liquid was about to overflow the copper mesh. After standing for approximately 20 min, the excess liquid was blotted with filter paper. Twenty microliters of 2% phosphotungstic acid (pH 5.52) were dropped onto the copper net and negatively stained for approximately 5 s. Then, filter paper was used to absorb any excess liquid. The sample was carefully placed into the sample box to make a record. The copper mesh was placed into the testing rod and shaken on a machine (JEOL, Akishima, Tokyo, Japan).

### 2.6. RNA Extraction and Reverse Transcription

RNA was extracted according to the manufacturer’s instructions using a Qiagen miRNeasy Micro Kit (No. 217084, Qiagen, Dusseldorf, North Rhine-Westphalia, Germany) and stored in a freezer at −80 °C. RNA reverse transcription was carried out using a PrimeScript™ RT Reagent Kit with gDNA Eraser (RR047A, Takara, Kyoto, Japan).

### 2.7. Real-Time Quantitative Polymerase Chain Reaction (qPCR)

An ABI 7500 fluorescent PCR instrument (GE, Boston, MA, USA) was used for real-time quantitative PCR detection. The primers and probes are listed in Appendix A. Twenty microliters of the qPCR reaction system were used according to the manufacturer’s instructions (RR390A, Takara, Kyoto, Japan). All the samples were performed in triplicate. Raw data were normalized and analyzed using the relative quantification method. ACTB was used as the internal control. The PCA3 or MALAT1 score was calculated as 2^−Ct(PCA3/MALAT1)−Ct(ACTB)^ × 1000.

### 2.8. Statistics

Statistical analysis was performed using R software v3.5.0, Med calc v13.0 (MedCalc Software bvba) and SPSS software v21.0 (IBM). Student’s *t*-test was used to compare age differences of the patients. The Mann–Whitney *U* test was used to compare the differences in tPSA, %fPSA and prostate volume of the patients. Pearson’s chi-squared test was used to evaluate the DRE status. The difference in the lncRNA scores between the patients with positive biopsies and the patients with negative biopsies was analyzed using the Mann-Whitney *U* test. Logistic regression analysis was used to establish a regression model combining the PCA3 score with the MALAT1 score. The lncRNA assay was based on the algorithm of the regression model. The ROC (receiver operating characteristic) curve and the area under the curve (AUC) were used to evaluate the diagnostic performance of the lncRNAs and the PSA. A DCA (decision curve analysis) was used to evaluate the net benefit for patients; *p* < 0.05 was considered statistically significant.

## 3. Results

### 3.1. Sample and Subject Characteristics

From January 2018 to April 2020, a total of 628 patients undergoing prostatic biopsy were enrolled in our study. Fifteen patients were excluded due to incomplete data. Another 48 patients were excluded due to insufficient exosomal RNA extraction. Therefore, 565 patients were finally recruited in this study.

The overall positive rate of the prostate biopsies was 38.1% (139/365) and 39.5% (79/200) in the discovery cohort and the validation cohort, respectively. The average age of the discovery cohort was 68.3 (SD: 7.9) years, and the average age of the validation cohort was 65.5 (SD: 7.2) years. The median serum PSA was 9.3 ng/mL (IQR: 6.7–12.9 ng/mL) and 8.9 ng/mL (IQR: 6.6–12.4 ng/mL), respectively. In the discovery cohort and the validation cohort, the prostate volume of patients with positive biopsy results was smaller than that of the negative result patients (58.5 vs. 43.2 and 44.6 vs. 27.7). PCa-associated risk factors, including total PSA (tPSA), volume and percent-free PSA (%fPSA), achieved statistical significance among the patients with a positive biopsy and those with a negative biopsy in the discovery cohort and the validation cohort. The demographics and clinical characteristics of the participants are presented in Table 1. We purified the exosomes from the urine samples according to the manufacturer’s instructions. The quality control of exosome isolation and verification is shown in Figure 1.

### 3.2. LncRNA Assay Predicted the Initial Biopsy Results

To investigate whether urinary exosomal PCA3 and MALAT1 could be potential biomarkers to detect PCa, we first compared the PCA3 and MALAT1 scores in PCa with those in patients with a negative prostate biopsy. The results showed that both the PCA3 and MALAT1 scores were significantly higher in the PCa group (Figure 2a, PCA3: *p* < 0.001; MALAT1: *p* < 0.001). In addition, both the PCA3 score and the MALAT1 score demonstrated a good performance in predicting the initial biopsy result, with areas under the receiver operating characteristic (ROC) curves of 0.751 and 0.791, respectively (Figure 2c). We then used a logistic regression analysis to construct an lncRNA assay by combining urinary exosomal PCA3 and MALAT1. The ROC analysis showed that the lncRNA assay yielded an AUC of 0.828, which was much better than that of PCA3 alone or MALAT1 alone (lncRNA assay vs. MALAT1 score, *p* = 0.032; lncRNA assay vs. PCA3 score, *p* < 0.001).

We also constructed a clinical base model by combining PCa-associated clinical features confirmed by the multivariable logistic regression to predict the initial biopsy results, and these factors included age, prostate volume, DRE status and f/tPSA (Table 2). After comparing the ROC curve of the lncRNA assay with those of the base model, PSA and f/t PSA, it was determined that the AUC value of the lncRNA assay was much higher than those of the PSA (*p* < 0.001), f/tPSA (*p* < 0.001) and the base model (*p* = 0.001) for distinguishing PCa from patients with a negative biopsy (Figure 2d and Table 2).

### 3.3. LncRNA Assay Predicted the Aggressiveness of PCa

To determine whether the PCA3 and MALAT1 scores were correlated with the tumor grade of PCa, we evaluated the PCA3 and MALAT1 scores in the clinically significant PCa (GS ≥ 7) with those in nonaggressive disease (PCa with GS6 and benign disease). The results showed that the expression levels of PCA3 and MALAT1 were both significantly increased in the clinically significant PCa group (GS ≥ 7) compared with the nonaggressive disease group (Figure 2b, PCA3: *p* < 0.001; MALAT1: *p* < 0.001). The AUCs of PCA3 and MALAT1 for diagnosis of clinically significant PCa were 0.723 and 0.806, respectively (Figure 2e).

The AUC analysis showed that the lncRNA assay combining the PCA3 score and the MALAT1 score yielded an AUC of 0.831, which was much higher than that of PCA3 (0.831 vs. 0.723, *p* < 0.001), to distinguish clinically significant PCa from nonaggressive diseases. In addition, the diagnostic performance of the lncRNA assay was superior to that of *MALAT1* alone but did not meet statistical significance (0.831 vs. 0.806, *p* = 0.137). We further compared the diagnostic value of the lncRNA assay with those of the PCa-associated clinical features and the base model including age, prostate volume and DRE status. The diagnostic power of the lncRNA assay was much better than that of the PSA (*p* < 0.001) model, fPSA/PSA (*p* < 0.001) and the base model (*p* < 0.001) in the diagnosis of clinically significant PCa (Table 2 and Figure 2f).

### 3.4. Validating the LncRNA Assay

To further validate the diagnostic performance of the lncRNA assay, we evaluated the parameters of the assay to predict PCa for the independent validation in a multicenter cohort of patients. Similarly, both the PCA3 and MALAT1 scores were significantly higher in the PCa group compared to those in the patients with a negative prostate biopsy (Figure 3a, PCA3: *p* < 0.001; MALAT1: *p* < 0.001) and in the PCa group compared to those in the patients with nonaggressive disease (Figure 3b, PCA3: *p* < 0.001; MALAT1: *p* < 0.001). In addition, both the PCA3 score and the MALAT1 score demonstrated a good performance in distinguishing PCa from controls (Figure 3c and Table 3) and clinically significant PCa from nonaggressive diseases (Figure 3e and Table 3). The lncRNA assay yielded an AUC of 0.814, which was superior to those of the PSA (*p* < 0.001), f/tPSA (*p* < 0.001) and the base model (*p* = 0.065) for distinguishing PCa from patients with a negative biopsy (Figure 3d and Table 3). The lncRNA assay also showed a better diagnostic performance than those of the PSA (*p* = 0.014), fPSA/PSA (*p* = 0.012) and the base model (0.779 vs. 0.719, *p* = 0.242) in the diagnosis of clinically significant PCa (Figure 3f and Table 3).

### 3.5. Clinical Utility of the lncRNA Assay

To assess the clinical value of the lncRNA assay, we first adopted a decision curve analysis (DCA) to evaluate the patients’ net benefit by comparing the results of the lncRNA assay with the results of the current clinical parameters. As the decision curve indicated, the lncRNA assay was superior to the PSA, f/tPSA and the base model for detecting PCa, with a higher net benefit for almost all threshold probabilities (Figure 4a). In addition, for differentiating clinically significant PCa from the nonaggressive disease group, the lncRNA assay also presented the highest net benefit across all threshold probabilities (Figure 4b). The distribution of biopsy results from the patients with the lncRNA assay score is depicted in a waterfall plot. The cutoff values were set according to the sensitivity of 90% and 95% to illustrate the distribution of biopsy results either above or below these cutoff values (Figure 4c). At the cutoff value of 90% sensitivity, the lncRNA assay could spare 30.9% (175) unnecessary biopsies while only missing 2.1% (12) of the cases of clinically significant PCa. When the cutoff value of 95% was applied, 24.2% (137) of unnecessary biopsies were prevented at the risk of missing only 7 (1.2%) cases of clinically significant PCa. These results demonstrated that the lncRNA assay had a significant clinical value in diagnosing PCa and clinically significant PCa compared to the current clinical parameters.

## 4. Discussion

With the increase in PSA screening, the diagnostic rate of prostate cancer has steadily increased [16]. While abnormal DRE and transrectal ultrasound (TRUS) results prompt a biopsy of the prostate in patients [17], the PSA is still the most reliable prostate biopsy indicator. As a routinely used test, higher levels of the PSA indicate a greater likelihood of PCa. However, the low specificity of the PSA testing for screening PCa and its limitation in detecting clinically significant PCa urge scientists to develop more specific biomarkers to diagnose clinically significant PCa early while avoiding unnecessary biopsies [18].

Previous reports demonstrated that PCA3 was present in urine and could be a diagnostic marker for prostate cancer [19,20,21]. A recent meta-analysis of PCA3 including 54 studies (17,575 patients) indicated that the overall ROC of PCA3 was 0.75 (95% CI: 0.71–0.79) [22]. In addition, the diagnostic performance of urinary exosomal PCA3 was also investigated [23,24]. Here, we showed that urinary exosomal PCA3 alone achieved an AUC of 0.751 in distinguishing PCa from the patients with a negative biopsy and an AUC of 0.732 in predicting clinically significant PCa. MALAT1 is another lncRNA that was first reported in lung cancer [25]. Our previous report demonstrated that urinary MALAT1 (AUC: 0.742) had a better diagnostic performance than the PSA (AUC: 0.601) and fPSA/PSA (AUC: 0.627) in patients with the PSA of 4–10 ng/ml [13]. Although urinary MALAT1 was identified as a potential biomarker of prostate cancer, its expression in urinary exosomes is still unclear. In this study, we showed that urinary exosomal MALAT1 had a good performance in predicting prostate biopsy results (AUC: 0. 791) and determining clinically significant PCa (AUC: 0.806). Therefore, our results indicated that both urinary exosomal PCA3 and MALAT1 could serve as promising biomarkers to detect PCa and clinically significant PCa.

As reported before, a combination of biomarkers [26], such as the MiPS (Mi Prostate Score Urine test), which contains serum PSA, urinary PCA3 and TMPRSS2-ERG (transmembrane protease, serine 2, v-ets erythroblastosis virus E26 oncogene like), could provide a higher accuracy for high-grade PCa than the PSA alone (AUC: 0.729 vs. 0.651, *p* < 0.001) [27]. Another urinary test called selectMDx, combined with two urinary mRNAs, HOXC6 (homeobox C6) and DLX1 (distal-less homeobox 1), had a high accuracy of predicting high-grade PCa [7,28]. Furthermore, another research developed an assay called ExoDx Prostate (IntelliScore) (EPI), which contained three urinary exosome genes (PCA3, *ERG* and *SPDEF* (SAM pointed domain-containing Ets transcription factor)) to improve the diagnosis of high-grade PCa (AUC: 0.77) and avoid unnecessary biopsies [29]. In our study, we found that the expression levels of the lncRNA assay were significantly increased in the patients with high Gleason scores (≥7) compared to the patients with low Gleason scores (Appendix A), indicating the lncRNA assay was corelated with the malignancy grade of PCa. Therefore, we constructed a urinary exosomal lncRNA assay consisting of PCA3 and MALAT1 to diagnose PCa and aggressive PCa. Our lncRNA assay achieved an AUC of 0.828 in predicting the biopsy results, which was superior to the current clinical parameters. More importantly, the lncRNA assay presented an excellent performance in distinguishing clinically significant PCa (AUC: 0.831), which was even better than the previous urinary exosomal assays [23,24]. For the clinical applications, the DCA and the waterfall plot demonstrated that the lncRNA assay had a higher net benefit in diagnosing PCa and clinically significant PCa than the current clinical parameters but could prevent a large number of unnecessary biopsies, indicating its significant clinical value. In addition, we managed to establish a comprehensive model which combined the lncRNA assay with clinical data (Appendix A). The new model showed an even better diagnostic performance and thus could be a useful promising tool for early diagnosis of PCa in the future.

Our study may provide new insight into the diagnosis of PCa and clinically significant PCa. However, there still are a few limitations. First, the sample size was limited. A further multicenter prospective large-scale study is needed to confirm our findings. Second, we did not perform head-to-head comparisons between the lncRNA assay and other emerging assays, including the MiPS, SelectMDx, EPI, etc. Finally, given the remarkable distinction of genetic alteration signatures between the Asian and Western populations [30], additional studies should be launched to compare the clinical value of our lncRNA assay in Asian patients compared to that in Western cohorts.

## 5. Conclusions

In conclusion, we developed and validated a novel noninvasive post-DRE urine-based lncRNA assay that presented good diagnostic power for the early diagnosis of PCa and high-grade PCa. Regarding clinical utility, the lncRNA assay had a higher net benefit than the current clinical parameters but could spare substantial unnecessary biopsies.

## Figures and Tables

**Figure 1 cancers-13-04075-f001:**
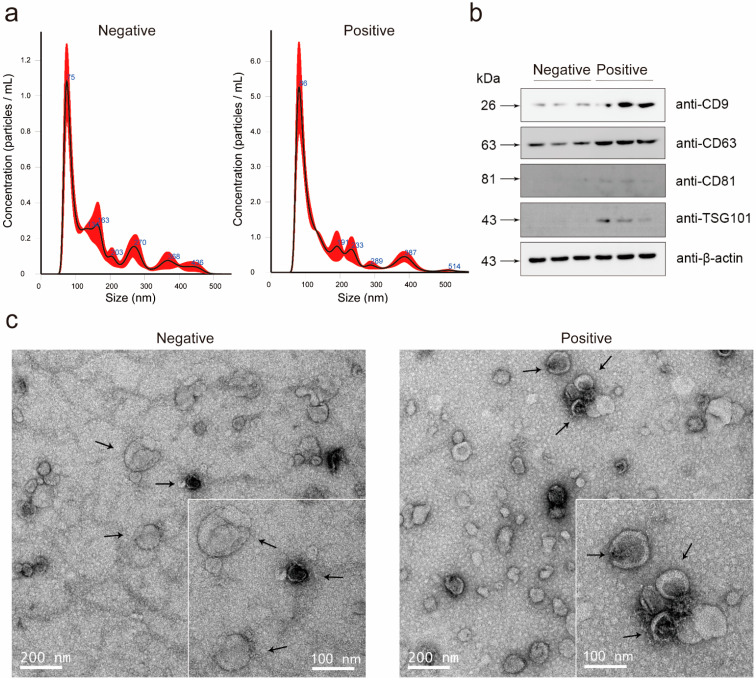
Characteristics of urinary exosomes. (**a**) Nanotracking analysis shows the size distribution of the urinary exosomes of the biopsy-negative and biopsy-positive patients. The particle size of the urine exosomes was concentrated at 40–160 nm in both the biopsy-negative and biopsy-positive patients. The particle size peak distribution of the urinary exosomes of the biopsy-negative and biopsy-positive patients was at 75 nm and 86 nm, respectively. (**b**) Analysis of the expression of the exosomal markers CD9, CD63, CD81, TSG101 and internal control ACTB using Western blots. (**c**) Transmission electron microscopy graphs show the morphology of the urine exosomes of the biopsy-negative and biopsy-positive patients. The arrows were used to point the representative exosomes.

**Figure 2 cancers-13-04075-f002:**
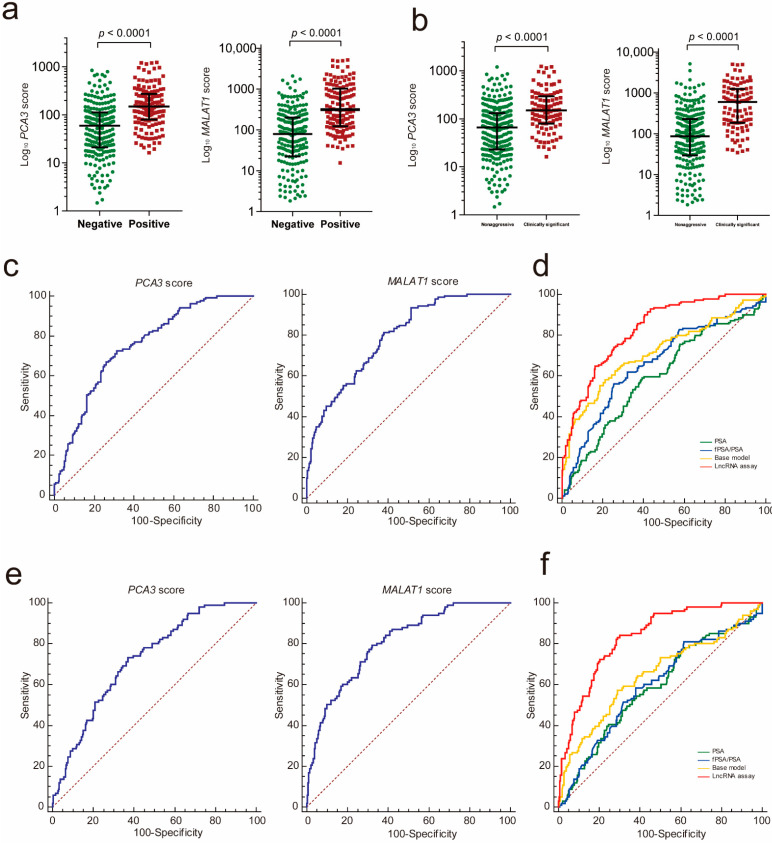
The lncRNA assay predicted the initial biopsy results. (**a**) Urinary exosomal PCA3 and MALAT1 were significantly increased in the PCa patients compared to the patients with a negative prostate biopsy (PCA3: *p* < 0.001; MALAT1: *p* < 0.001). (**b**) Urinary exosomal PCA3 and MALAT1 were significantly increased in the clinically significant PCa patients compared with the patients with nonaggressive disease (PCA3: *p* < 0.001; MALAT1: *p* < 0.001). The ROC analysis shows the diagnostic power of the PCA3 score and the MALAT1 score in PCa (**c**) and clinically significant PCa (**e**). Comparison ROC analysis shows that the lncRNA assay has a better diagnostic performance than fPSA/PSA, PSA and the base model in PCa (**d**) and clinically significant PCa diagnosis (**f**). ROC—receiver operating characteristic; PCA3—prostate cancer antigen 3; MALAT1—metastasis-associated lung adenocarcinoma transcript 1.

**Figure 3 cancers-13-04075-f003:**
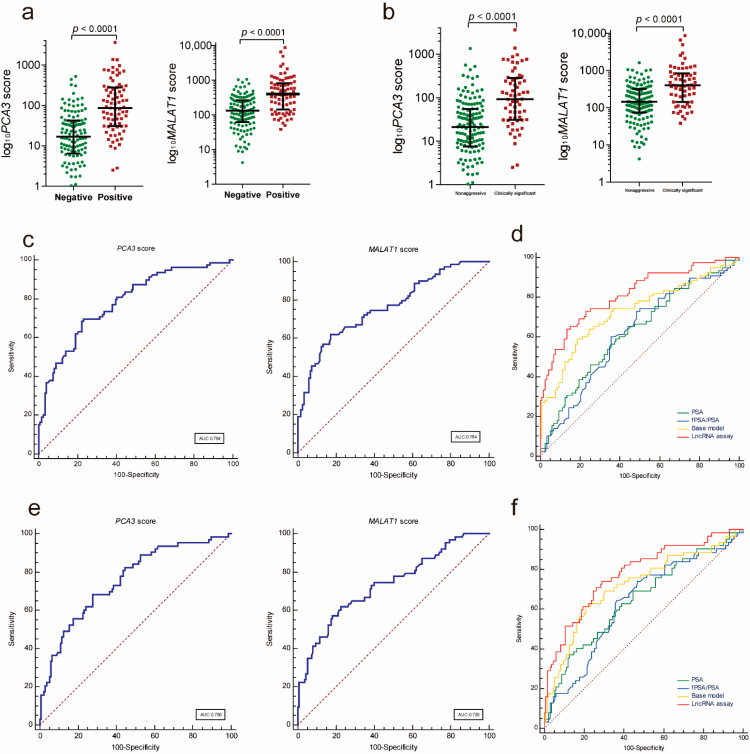
Diagnostic performance of the lncRNA assay in the validation cohort. (**a**) Urinary exosomal PCA3 and MALAT1 were significantly increased in the PCa patients compared to the patients with a negative prostate biopsy (PCA3: *p* < 0.001; MALAT1: *p* < 0.001). (**b**) Urinary exosomal PCA3 and MALAT1 were significantly increased in the clinically significant PCa patients compared with the patients with nonaggressive disease (PCA3: *p* < 0.001; MALAT1: *p* < 0.001). The ROC analysis shows the diagnostic power of the PCA3 score and the MALAT1 score in diagnosing PCa ((**c**), AUC: 0.784 and 0.764, respectively) and clinically significant PCa ((**e**), AUC: 0.756 and 0.735, respectively). Comparison ROC analysis shows that the lncRNA assay has a better diagnostic performance than the fPSA/PSA, PSA and the base model in diagnosing PCa ((**d**), AUC: 0.814) and clinically significant PCa ((**f**), AUC: 0.779). ROC—receiver operating characteristic; PCA3—prostate cancer antigen 3; MALAT1—metastasis-associated lung adenocarcinoma transcript 1.

**Figure 4 cancers-13-04075-f004:**
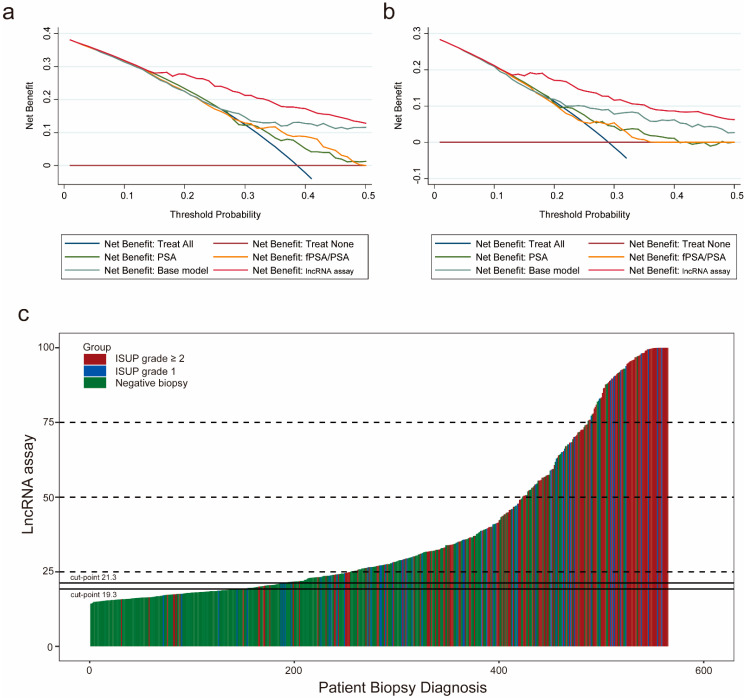
Clinical utility of the lncRNA assay. The decision curve analysis shows that the lncRNA assay presents the highest net benefit across all threshold probabilities for diagnosing PCa (**a**) and high-grade PCa (**b**). The dark red horizontal line parallel to the x-axis represents no patient undergoing a biopsy (Treat None). The blue line indicates that all the patients will have PCa (Treat All). (**c**) Waterfall plot of the lncRNA assay scores in relation to the prostate biopsy results (*n* = 565). Each bar represents an individual. Red indicates the ISUP grade ≥ two tumors (GS ≥ 7); blue indicates the ISUP grade of one tumor (GS = 6); green indicates the negative biopsies. Two black horizontal lines represent the cutoff points of 21.3 at the sensitivity of 90% and 19.3 at the sensitivity of 95%. ISUP: International Society of Urological Pathology.

**Table 1 cancers-13-04075-t001:** Demographics and clinical characteristics of participants.

Parameter	Discovery Cohort				Validation Cohort			
	Entire	Negative	Positive	*p*-Value	Entire	Negative	Positive	*p*-Value
Age, years				0.03 *				0.052 *
Number of patients (%)	365 (100.0)	226 (61.9)	139 (38.1)		200 (100.0)	121 (60.5)	79 (39.5)	
Mean	68.3	67.6	69.4		65.5	64.7	66.8	
SD	7.9	8.2	7.5		7.2	7.5	6.8	
tPSA, ng/mL				0.002 ^#^				0.004 ^#^
Number of patients (%)	365 (100.0)	226 (61.9)	139 (38.1)		200(100.0)	121(60.5)	79(39.5)	
Median	9.3	8.8	10.5		8.9	8.4	9.9	
IQR	6.7–12.9	6.5–12.1	7.8–13.6		6.6–12.4	6.4–11.1	7.5–13.5	
Volume, mL				<0.001 ^#^				<0.001 ^#^
Number of patients (%)	365 (100.0)	226 (61.9)	139 (38.1)		200 (100.0)	121 (60.5)	79 (39.5)	
Median	52.6	58.5	43.2		41	44.6	27.7	
IQR	35.8–73.9	41.4–75.7	27.9–68.9		25.6–59.9	31.4–62.5	19.9–57.0	
%fPSA				<0.001 ^#^				0.004 ^#^
Number of patients (%)	365 (100.0)	226 (61.9)	139 (38.1)		196 (98.0)	118 (0.59)	78 (0.39)	
Median	0.16	0.18	0.13		0.14	0.15	0.12	
IQR	0.11–0.22	0.13–0.23	0.09–0.18		0.09–0.18	0.10–0.20	0.09–0.16	
Suspicious DRE				<0.001 ^§^				0.313 ^§^
Number of patients	365 (100.0)	226 (61.9)	139 (38.1)		200 (100.0)	117 (58.5)	74 (37)	
No. (%)	105 (28.8)	45 (12.3)	60 (16.4)		9 (4.5)	4 (2)	5 (2.5)	
Biopsy Gleason score (%)								
6			38 (10.4)				16 (8)	
7			54 (14.8)				36 (18)	
≥8			47 (12.8)				26 (13)	

PSA = prostate-specific antigen; SD = standard deviation; IQR = interquartile range; tPSA = total PSA; %fPSA = percent-free PSA; DRE = digital rectal examination. Estimated by transrectal ultrasound. * Student’s *t*-test. ^#^ Mann–Whitney *U* test. ^§^ Pearson’s chi-squared test.

**Table 2 cancers-13-04075-t002:** Performance of the lncRNA assay and PCa-associated clinical features to predict biopsy results in the discovery cohort.

Parameters	Positive and Negative			Nonaggressive and Clinically Significant		
	AUC	Univariate *p*	Multivariate *p*	AUC	Univariate *p*	Multivariate *p*
Age	0.568	0.026	0.006	0.566	0.049	0.043
BMI	0.520	0.530	0.389	0.539	0.255	0.129
DRE	0.616	<0.001	<0.001	0.589	0.002	0.008
Volume	0.627	0.002	0.012	0.620	<0.001	0.033
PSA	0.596	0.002	0.131	0.586	0.011	0.111
fPSA/PSA	0.661	<0.001	<0.001	0.596	0.005	0.190
PCA3	0.751	<0.001	<0.001	0.723	<0.001	<0.001
MALAT1	0.791	<0.001	<0.001	0.806	<0.001	<0.001
LncRNA assay	0.828	<0.001	-	0.831	<0.001	-

OR: odds ratio; CI: confidence interval; AUC: area under the curve; PSA: prostate-specific antigen; DRE: digital rectal examination; fPSA: free prostate-specific antigen; PCA3 (prostate cancer antigen 3); MALAT1 (metastasis-associated lung adenocarcinoma transcript 1). The base model of positive and negative biopsy diagnosis consists of age, volume, fPSA/PSA and DRE. The base model of nonaggressive disease and clinically significant PCa consists of age, DRE and volume.

**Table 3 cancers-13-04075-t003:** Performance of the lncRNA assay and PCa-associated clinical features in predicting biopsy results in the validation cohort.

Parameters	Positive and Negative			Nonaggressive and Clinically Significant		
	AUC	Univariate *p*	Multivariate *p*	AUC	Univariate *p*	Multivariate *p*
Age	0.563	0.049	0.027	0.543	0.139	0.197
BMI	0.608	0.011	0.036	0.596	0.033	0.083
DRE	0.515	0.319	0.635	0.525	0.127	0.576
Volume	0.649	0.008	0.007	0.643	0.021	0.028
PSA	0.620	0.005	0.006	0.642	<0.001	0.002
fPSA/PSA	0.619	0.106	0.400	0.631	0.133	0.440
PCA3	0.784	<0.001	<0.001	0.756	<0.001	0.010
MALAT1	0.764	<0.001	<0.001	0.735	<0.001	0.001
LncRNA assay	0.814	<0.001	-	0.781	<0.001	-

The base model of positive and negative biopsy diagnosis consists of age, BMI, volume, PSA. The base model of the nonaggressive disease and clinically significant PCa consists of BMI, volume, PSA.

## Data Availability

All the data associated with this study are presented in the article. Correspondence and requests for materials should be addressed to F.W. (wangbofengye@163.com).

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
