# Peer review of "A Novel Urine Exosomal lncRNA Assay to Improve the Detection of Prostate Cancer at Initial Biopsy: A Retrospective Multicenter Diagnostic Feasibility Study"

_cancers, 2021, doi:10.3390/cancers13164075_

Round 1

Reviewer 1 Report

The authors have addressed all the points raised and the manuscript is now ready for the publication. 

Author Response

Thank you. We are glad that the you are satisfied with our revision.

Reviewer 2 Report

none

Author Response

(The authors gave the same response as above.)

Reviewer 3 Report

Because of the low numbers of small study population,   the authors should include the term "feasibility study" 

Author Response

Thanks for your constructive comment. we changed title to:A novel urine exosomal lncRNA assay to improve the detection of prostate cancer at initial biopsy: a retrospective, multicenter, diagnostic, feasibility study

Reviewer 4 Report

I appreciate the authors for their detailed explanations. Following is my additional comment.   -What is the "lncRNA assay"? The authors describe, "We then used a logistic regression analysis to construct a lncRNA assay by combining urinary exosomal PCA3 and MALAT1." in the paragraph of "LncRNA assay predicted the initial biopsy results." I think the details of the lncRNA assay are hard to understand from this description. Is the lncRNA the sum of the PCA3 score and MALAT1 score? The score of the lncRNA assay is a core point in this study; therefore, the authors need to describe what the lncRNA assay is or how to calculate the lncRNA assay score accurately. If the authors have already reported accurate information about the lncRNA assay, please let me know where the sentences are.

Author Response

Thanks for your constructive comment. We changed description of lncRNA assay in method section :  Logistic regression analysis was used to establish a regression model combining PCA3 score with MALAT1 score. The lncRNA assay was based on the algorithm of the regression model.

This manuscript is a resubmission of an earlier submission. The following is a list of the peer review reports and author responses from that submission.

Round 1

Reviewer 1 Report

Summary

 In this study, Yun Li et al. developed a non-invasive, urine exosome-based lncRNA assay for diagnosis of prostate cancer. The authors isolated the exosomes and detected the lncRNAs PCA3 and MALAT1 expression in prostate cancer patients. They concluded that combining the expression of PCA3 and MALAT1 in exosomes could be used as a potential biomarker for diagnosis of prostate cancer.  The study is interesting in terms of prostate cancer diagnosis and application, but needs further investigation to validate their hypothesis.

Comments:

  1. In Figure 2 and 3, the authors showed that lncRNA assay is good for predicting the initial prostate cancer biopsy result. In order to find the clinical relevance for lncRNA in predicting prognostic biomarkers, the authors should validate their findings with an independent prostate cancer data set.
  2. The patients with prostate cancer often undergo androgen deprivation therapy. I am wondering whether the PCA3 and MALAT1 expression will still be a good diagnostic marker in androgen depletion patients derived exosomes.
  3. The authors mentioned that expression of PCA3 and MALAT1 in prostate cancer-derived exosomes could be used as a good indicator for prostate cancer diagnosis. The authors should test whether the expression of PCA3 and MALAT1 in the exosomes correlates with prostate cancer, Gleason score for the diagnosis of prostate cancer.
  4. The authors should correct the spelling of the methodology part completely.
  5. Figure 3A is not visible and the authors should replace that Figure.

Author Response

(Reviewer 1)

In this study, Yun Li et al. developed a non-invasive, urine exosome-based lncRNA assay for diagnosis of prostate cancer. The authors isolated the exosomes and detected the lncRNAs PCA3 and MALAT1 expression in prostate cancer patients. They concluded that combining the expression of PCA3 and MALAT1 in exosomes could be used as a potential biomarker for diagnosis of prostate cancer.  The study is interesting in terms of prostate cancer diagnosis and application, but needs further investigation to validate their hypothesis.

Comments:

  1. In Figure 2 and 3, the authors showed that lncRNA assay is good for predicting the initial prostate cancer biopsy result. In order to find the clinical relevance for lncRNA in predicting prognostic biomarkers, the authors should validate their findings with an independent prostate cancer data set.

Response:Thanks for your constructive comment. We further validated our findings in a multi-center cohort of patients for the independent validation. We added the content of “Validating the lncRNA assay” in the section of Results.

For your review, the added information was also attached below.

Validating the lncRNA assay

To further validate the diagnostic performance of the lncRNA assay, we evaluated the parameters of the assay to predict PCa for the independent validation in a multi-center cohort of patients. Similarly, both the PCA3 and MALAT1 scores were significantly higher in the PCa group compared to those in patients with a negative prostate biopsy (Figure 3a, PCA3: P<0.001; MALAT1: P<0.001), and in the PCa group compared to those in patients with the non-aggressive diseases (Figure 3b, PCA3: P<0.001; MALAT1: P<0.001). In addition, both the PCA3 score and the MALAT1 score demonstrated a good performance in predicting PCa from controls (Figure 3c and Table 3) and clinically significant PCa from non-aggressive diseases (Figure 3e and Table 3). The lncRNA assay yielded an AUC of 0.814, which was superior than those of PSA (P<0.001), f/tPSA (P<0.001) and the base model (P=0.065) for distinguishing PCa from patients with a negative biopsy (Figure 3d and Table 3). The lncRNA assay also showed a better diagnostic performance than those of PSA (P=0.014), fPSA/PSA (P=0.012), and the base model (0.779 vs 0.719, P=0.242) in the diagnosis of clinically significant PCa (Figure 3f and Table 3).

Figure 3

Figure 3 The Diagnostic performance of the lncRNA assay in validation cohort. a, Urinary exosomal PCA3 and MALAT1 were significantly increased in PCa patients compared to patients with a negative prostate biopsy (PCA3: P<0.001; MALAT1: P<0.001). b, Urinary exosomal PCA3 and MALAT1 were significantly increased in clinically significant PCa patients compared with patients with nonaggressive disease (PCA3: P<0.001; MALAT1: P<0.001). The ROC analysis shows the diagnostic power of the PCA3 score), MALAT1 score in diagnosing PCa (c, AUC: 0.784 and 0.764, respectively) and clinically significant PCa (e, AUC: 0.756 and 0.735, respectively). Comparison ROC analysis shows that the lncRNA assay has better diagnostic performance than fPSA/PSA, PSA, and the base model in diagnosing PCa (d, AUC:0.814) and clinically significant PCa (f, AUC: 0.779). ROC (receiver operating characteristic); PCA3 (prostate cancer antigen 3); MALAT1 (metastasis-associated lung adenocarcinoma transcript 1)

  1. The patients with prostate cancer often undergo androgen deprivation therapy. I am wondering whether the PCA3 and MALAT1 expression will still be a good diagnostic marker in androgen depletion patients derived exosomes.

Response:

Thanks for your comment. The clinical samples selected in this study were all treated naive for androgen deprivation therapy. The primary objective of this study was to evaluate the role of exosome biomarkers in the diagnosis of prostate cancer and clinically significant prostate cancer. We appreciated your suggestion. We will continue to pay attention to the PCa patients and their follow-up data in subsequent studies to judge the predictive value of the lncRNA assay on androgen deprivation therapy.

  1. The authors mentioned that expression of PCA3 and MALAT1 in prostate cancer-derived exosomes could be used as a good indicator for prostate cancer diagnosis. The authors should test whether the expression of PCA3 and MALAT1 in the exosomes correlates with prostate cancer, Gleason score for the diagnosis of prostate cancer.

Response:

Thank for your valuable comment. We analyzed the relationship between Gleason score and the lncRNA assay. We found that the expression levels of lncRNA assay were significantly increased in patients with high Gleason Scores (≥7) compared to patients with low Gleason Scores, indicating the lncRNA assay was corelated with the malignancy grade of PCa. We added this information in the section of Discussion.

For your review, the added information was also attached below.

“In our study, we found that the expression levels of lncRNA assay were significantly increased in patients with high Gleason Scores (≥7) compared to patients with low Gleason Scores (Supplementary Figure 1), indicating the lncRNA assay was corelated with the malignancy grade of PCa.”

Supplementary Figure 1

Supplementary Figure 1

 The lncRNA assay was corelated with the malignancy grade of prostate cancer (PCa)

  1. The authors should correct the spelling of the methodology part completely.

Thanks for your valuable comment. Written English of our manuscript including the methodology part has been extensively edited by American Journal Experts (AJE) with a verification code AJE: A7B6-2A4F-FDCC-A0A0-B9D6.

The certificate attached below.

  1. Figure 3A is not visible and the authors should replace that Figure.

Response:

Thank for your comment. We have changed the picture with high resolution.

Reviewer 2 Report

In this manuscript Yun Li et al investigate whether urinary exosomal PCA3 and MALAT1 could function as potential diagnostic biomarkers PCa. They present data on the development and validation of this non-invasive, post-DRE lncRNA assay, which support its clinical utility for the early diagnosis of PCa and high-grade PCa.

Major points.

(A) In Figures 2a and 2b the authors compare PCA3 and MALAT1 scores in PCa patients with a positive vs negative prostate biopsies and in PCa patients with significant PCa (GS≥7) vs non-aggressive disease, respectively. The results showed that both, PCA3 190 and MALAT1 scores were significantly higher in patients with positive biopsies as opposed to those with negative ones; in addition, higher scores for both markers could also be detected in patients with aggressive disease vs those with non-aggressive disease. In these figures the authors show individual values and means. The problem is that the vast majority of individual values in the positive and clinically significant groups (in which PCA3 and MALAT1 have higher mean values (SDs are missing)) are within the range of values scored in the PCa negative group (Fig.2a) and the PCa non-aggressive group (Fig.2b). This implies that the proposed lncRNA assay should be always used along with the established diagnostic ones for PCa. If, so, then which would be its distinctive value?

(B) The authors show the distribution of biopsies from patients with the lncRNA assay score in a waterfall plot. Accordingly, at the cut-off value of 90% sensitivity, the lncRNA assay missed 7.7% (n=28) clinically significant PCa cases whereas at the cut-off value of 95%, 16 such cases were missed (4.3%). Although, the percentage of cases missed is low still these results point to the need to combine this novel assay with the ones already used in clinical praxis. Moreover, the waterfall plot additionally shows that there is a number of negative biopsy cases (or of low significance) which are included within the positive range of the lncRNA assay. How this should be interpreted?

Based on these points, I strongly recommend that the authors should validate their data in a larger cohort of PCa patients using the same stratifications.

In this manuscript Yun Li et al investigate whether urinary exosomal PCA3 and MALAT1 could function as potential diagnostic biomarkers PCa. They present data on the development and validation of this non-invasive, post-DRE lncRNA assay, which support its clinical utility for the early diagnosis of PCa and high-grade PCa.

Major points.

(A) In Figures 2a and 2b the authors compare PCA3 and MALAT1 scores in PCa patients with a positive vs negative prostate biopsies and in PCa patients with significant PCa (GS≥7) vs non-aggressive disease, respectively. The results showed that both, PCA3 190 and MALAT1 scores were significantly higher in patients with positive biopsies as opposed to those with negative ones; in addition, higher scores for both markers could also be detected in patients with aggressive disease vs those with non-aggressive disease. In these figures the authors show individual values and means. The problem is that the vast majority of individual values in the positive and clinically significant groups (in which PCA3 and MALAT1 have higher mean values (SDs are missing)) are within the range of values scored in the PCa negative group (Fig.2a) and the PCa non-aggressive group (Fig.2b). This implies that the proposed lncRNA assay should be always used along with the established diagnostic ones for PCa. If, so, then which would be its distinctive value?

(B) The authors show the distribution of biopsies from patients with the lncRNA assay score in a waterfall plot. Accordingly, at the cut-off value of 90% sensitivity, the lncRNA assay missed 7.7% (n=28) clinically significant PCa cases whereas at the cut-off value of 95%, 16 such cases were missed (4.3%). Although, the percentage of cases missed is low still these results point to the need to combine this novel assay with the ones already used in clinical praxis. Moreover, the waterfall plot additionally shows that there is a number of negative biopsy cases (or of low significance) which are included within the positive range of the lncRNA assay. How this should be interpreted?

Based on these points, I strongly recommend that the authors should validate their data in a larger cohort of PCa patients using the same stratifications.

Author Response

 (Reviewer 2)

In this manuscript Yun Li et al investigate whether urinary exosomal PCA3 and MALAT1 could function as potential diagnostic biomarkers PCa. They present data on the development and validation of this non-invasive, post-DRE lncRNA assay, which support its clinical utility for the early diagnosis of PCa and high-grade PCa.

Major points.

  • In Figures 2a and 2b the authors compare PCA3 and MALAT1 scores in PCa patients with a positive vs negative prostate biopsies and in PCa patients with significant PCa (GS≥7) vs non-aggressive disease, respectively. The results showed that both, PCA3 190 and MALAT1 scores were significantly higher in patients with positive biopsies as opposed to those with negative ones; in addition, higher scores for both markers could also be detected in patients with aggressive disease vs those with non-aggressive disease. In these figures the authors show individual values and means. The problem is that the vast majority of individual values in the positive and clinically significant groups (in which PCA3and MALAT1 have higher mean values (SDs are missing)) are within the range of values scored in the PCa negative group (Fig.2a) and the PCa non-aggressive group (Fig.2b). This implies that the proposed lncRNA assay should be always used along with the established diagnostic ones for PCa. If, so, then which would be its distinctive value?

Response:

Thank for your valuable comments. Firstly, in this study, we showed that urinary exosomal lncRNA was significantly increased in PCa compared to controls and in clinically significant PCa patients compared to non-aggressive disease patients (Figure 2A and 2B). The range of urine exosomal lncRNA of PCa patients were also significant elevated (P<0.001). Generally, a very few biomarkers that are only highly expressed in cancer and not expressed in non-cancer, or the expression range is much lower than that of cancer. Although there will always be some overlaps in the expression range, the overall expression levels of tumor biomarkers in the cancer group are much higher than those in the non-cancerous group. And this phenomenon does not affect the diagnostic performance of such biomarkers, like PSA, CEA, AFP etc, which were widely used in clinic nowadays, as well as the novel biomarkers, like the lncRNA assay and other novel ones. Auprich, M1.et al reported that PCA3 was highly expressed in prostate cancer tissues relative to benign tissues. The expression range between cancers and controls was also overlapping. Bernard, V2. et al reported that the circulating nucleic acids were associated with outcomes of patients with pancreatic cancer including exosomes’ DNA and ctDNA. Also, the expression range was overlapping. The expression of the lncRNA assay is similar to the results of these above studies. The median and interquartile range were covered by the points with the same colour. We changed the colour of the figure.

Secondly,We greatly appreciate your thoughtful and constructive comments. We compared lncRNA assay with the existing base model (consisting of clinical parameters) and found that its diagnostic effect was also significantly improved, indicating that the new biomarkers are better than the existing markers, which also shows the independent diagnostic value of the lncRNA assay (Figure2d and 2f).

Thirdly, we strongly agree with Reviewer's opinion that we need to combine our biomarkers with clinical data to establish a comprehensive model. As the scatter plot shows, the mean values of prostate cancer and cs prostate cancer are significantly higher than the control (P<0.0001). The AUC of comprehensive moedel (Basemodel+lncRNA assay) is 0.846 for cancer detection. The AUC of comprehensive model (Basemodel+lncRNA assay) is 0.811 for clinically significant prostate cancer detection.

For your review, the added information was also attached below. The following sentences were added in the section of Discussion.

“In addition, we managed to establish a comprehensive model which combined the lncRNA assay with clinical data (Supplementary Figure 2). The new model showed an even better diagnostic performance and thus could be a useful promising tool for early diagnosis of PCa in the future.”

The Supplementary Figure 2 was attached below:

Supplementary Figure 2 The diagnosis performance of comprehensive model. a, The probability were significantly elevated in PCa patients than negative biopsy patients. b, The probability were significantly elevated in clinically significant PCa patients than non-aggressive disease patients. ROC analysis shows the diagnostic power of comprehensive model in PCa (c, AUC:0.846) and clinically significant PCa (e, AUC:0.811).

  • The authors show the distribution of biopsies from patients with the lncRNA assay score in a waterfall plot. Accordingly, at the cut-off value of 90% sensitivity, the lncRNA assay missed 7.7% (n=28) clinically significant PCa cases whereas at the cut-off value of 95%, 16 such cases were missed (4.3%). Although, the percentage of cases missed is low still these results point to the need to combine this novel assay with the ones already used in clinical praxis. Moreover, the waterfall plot additionally shows that there is a number of negative biopsy cases (or of low significance) which are included within the positive range of the lncRNA assay. How this should be interpreted?

Response:

Thank for your constructive comments. We used waterfall plot to depict the distribution of every participant with the corresponding expression levels of the lncRNA assay. As shown in Figure 4c, readers could clearly distinguish clinically significant prostate cancer from non-aggressive diseases including patients with ISUP grade 1 and negative biopsies. In addition, we agree with the reviewer that using existing lncRNA assays will still miss a small number of clinically significant prostate cancers. For now, we still have a long way to go to identify the perfect biomarkers which could detect all aggressive cancers without unnecessary prostate biopsies. Here are strategies we will follow to deal with this issue. Firstly, we will conduct a multicenter, prospective, large-scale study to improve the cutoff values of the assay. Secondly, we will combine the assay with all available clinical data to establish a comprehensive model to improve the detection of PCa and clinically significant PCa. Thirdly, we can repeat the lncRNA assay and comprehensive model in the clinic to avoid the false negative cases.

Based on these points, I strongly recommend that the authors should validate their data in a larger cohort of PCa patients using the same stratifications.

Response:

Response:Thanks for your constructive comment. We further validated our findings in a multi-center cohort of patients for the independent validation. We added the content of “Validating the lncRNA assay” in the section of Results.

For your review, the added information was also attached below.

Validating the lncRNA assay

To further validate the diagnostic performance of the lncRNA assay, we evaluated the parameters of the assay to predict PCa for the independent validation in a multi-center cohort of patients. Similarly, both the PCA3 and MALAT1 scores were significantly higher in the PCa group compared to those in patients with a negative prostate biopsy (Figure 3a, PCA3: P<0.001; MALAT1: P<0.001), and in the PCa group compared to those in patients with the non-aggressive diseases (Figure 3b, PCA3: P<0.001; MALAT1: P<0.001). In addition, both the PCA3 score and the MALAT1 score demonstrated a good performance in predicting PCa from controls (Figure 3c and Table 3) and clinically significant PCa from non-aggressive diseases (Figure 3e and Table 3). The lncRNA assay yielded an AUC of 0.814, which was superior than those of PSA (P<0.001), f/tPSA (P<0.001) and the base model (P=0.065) for distinguishing PCa from patients with a negative biopsy (Figure 3d and Table 3). The lncRNA assay also showed a better diagnostic performance than those of PSA (P=0.014), fPSA/PSA (P=0.012), and the base model (0.779 vs 0.719, P=0.242) in the diagnosis of clinically significant PCa (Figure 3f and Table 3).

Figure 3 The Diagnostic performance of the lncRNA assay in invalidation cohort. a, Urinary exosomal PCA3 and MALAT1 were significantly increased in PCa patients compared to patients with a negative prostate biopsy (PCA3: P<0.001; MALAT1: P<0.001). b, Urinary exosomal PCA3 and MALAT1 were significantly increased in clinically significant PCa patients compared with patients with nonaggressive disease (PCA3: P<0.001; MALAT1: P<0.001). The ROC analysis shows the diagnostic power of the PCA3 score), MALAT1 score in PCa (c, AUC: 0.784 and 0.764, respectively) and clinically significant PCa (e, AUC: 0.756 and 0.735, respectively). Comparison ROC analysis shows that the lncRNA assay has better diagnostic performance than fPSA/PSA, PSA, and the base model in PCa (d, AUC:0.814) and clinically significant PCa diagnosis (f, AUC: 0.779). ROC (receiver operating characteristic); PCA3 (prostate cancer antigen 3); MALAT1 (metastasis-associated lung adenocarcinoma transcript 1)

Reviewer 3 Report

The authors have submitted a MS describing  non-invasive a  lncRNA-based  assay aimed ad  the early diagnosis of Prostate Ca

The MS is clear and the results properly presented. As the authors themselves admit, the study population size represents  a significant limitation of the study. Therefore I recommend the authors to refer to  their study from the title onwards as a "feasibility study"

Abbreviation should be explained from the first use throughout the text         

Author Response

The authors have submitted a MS describing  non-invasive a  lncRNA-based  assay aimed ad  the early diagnosis of Prostate Ca

The MS is clear and the results properly presented. As the authors themselves admit, the study population size represents  a significant limitation of the study. Therefore I recommend the authors to refer to  their study from the title onwards as a "feasibility study"

Response:

Thank for your comment. We are glad that the you are satisfied with our paper. We modified the title “A novel urine exosomal lncRNA assay to improve the detection of prostate cancer at initial biopsy: a retrospective, multicenter, diagnostic study” according to your suggestion.

Abbreviation should be explained from the first use throughout the text   

Response:

Thank for your comment. We revised the manuscript We revised the manuscript according to the author's instructions.

Reviewer 4 Report

This manuscript reported the clinical effectiveness of lncRNA assay, PCA3, and MALAT1, to improve the detection of prostate cancer. I have some comments.  

- Authors used a lot of abbreviations and acronyms without explanations. Please follow the author's instruction manual listed below.   "Acronyms/Abbreviations/Initialisms should be defined the first time they appear in each of three sections: the abstract; the main text; the first figure or table. When defined for the first time, the acronym/abbreviation/initialism should be added in parentheses after the written-out form."  

Introduction - What is the author's most debatable issue or originality in this manuscript? The PCA3 assay has already been approved in some countries to reduce non-beneficial repeat prostate biopsies, and there have been many articles about PCA3. The authors write, "Finally, the present study was designed to establish a lncRNA assay to detect PCa and clinical signigicant PCa at initial prostate biopsy." But, I think only the biopsy can detect PCa. Although the PCA3 assay may indicate the presence of PCa, it cannot detect PCa. The initial prostate biopsy is a very important process in prostate cancer diagnosis. If the authors consider the PCA3 assays can replace or reduce the initial prostate biopsy, more logical explanations should be described.

- In the third paragraph, the word lncRNA PCA3 and MALAT1, was abruptly used. Many readers would not know what these RNAs are. The authors should describe why they focus on these lncRNAs.  

Materials and Methods -Is the biopsy protocol unified among study institutions? Differences in biopsy counts will lead to differences in cancer positive rate. Subcategory titles, WB, NTA, TEM, and qPCR, should be written with a full version. - Authors performed the qPCR method in this study. Please explain how to perform the reverse transcription. - In paragraph 2.7. qPCR, the authors state, "The PCA3 or MALAT1 score was calculated as 2-Ct(PCA3/MALAT1-Ct(ACTB)*1000." The closing parenthesis corresponding to the first parenthesis is missing. - In paragraph 2.8. Statistics, the authors state statistical methods only for the lncRNA score, the regression model of LncRNA assay, the ROC curve, and the DCA curve. The authors should describe all statistical methods using in this study. (e.g., statistical analyses performed for clinical characteristics of participants.)  

Results -The authors reported that the PCA3 score and the MALAT1 score in the cancer positive group/non-aggressive group were significantly higher than that in the cancer negative group/clinically significant group, but the specific figures are not shown in the text and the figure. Please explain the specific figures. -The authors analyze the effectiveness of the PCA3 score and the MALAT1 score by the ROC curve. Please indicate the cut-off values for the PCA3 score and the MALAT1 score, and describe why the authors select that cut-off value as the best optimal. According to Figure 2a and b, the PCA3 score and the MALAT1 score seems to have a vast range.  

Discussions -The authors should describe the limitations of this study.

Author Response

This manuscript reported the clinical effectiveness of lncRNA assay, PCA3, and MALAT1, to improve the detection of prostate cancer. I have some comments.  

- Authors used a lot of abbreviations and acronyms without explanations. Please follow the author's instruction manual listed below.   "Acronyms/Abbreviations/Initialisms should be defined the first time they appear in each of three sections: the abstract; the main text; the first figure or table. When defined for the first time, the acronym/abbreviation/initialism should be added in parentheses after the written-out form."

Response:

Thank for your valuable comment. We revised the manuscript according to the author's instructions.

Introduction - What is the author's most debatable issue or originality in this manuscript? The PCA3 assay has already been approved in some countries to reduce non-beneficial repeat prostate biopsies, and there have been many articles about PCA3. The authors write, "Finally, the present study was designed to establish a lncRNA assay to detect PCa and clinical signigicant PCa at initial prostate biopsy." But, I think only the biopsy can detect PCa. Although the PCA3 assay may indicate the presence of PCa, it cannot detect PCa. The initial prostate biopsy is a very important process in prostate cancer diagnosis. If the authors consider the PCA3 assays can replace or reduce the initial prostate biopsy, more logical explanations should be described.

Response:

Thanks for your constructive comment. To the best of our knowledge, there is no urinary exosomal lncRNA assay for the diagnosis of prostate cancer. We made it clear in the section of introduction that the aim of the study is to establish a urinary exosomal lncRNA assay for the early diagnosis of prostate cancer.

We agree with the reviewer that only biopsy can detect prostate cancer. We may not express our views clearly in the manuscript. The main purpose of this study is to use lncRNA assay to guide whether patients with suspected prostate cancer should undergo prostate biopsy, so as to increase the positive rate of prostate biopsy for prostate cancer and clinically significant prostate cancer and thus to avoid unnecessary biopsy. The main purpose of this study is to establish the lncRNA assay to improve existing diagnostic methods to guide the biopsy rather than directly diagnosing prostate cancer.

- In the third paragraph, the word lncRNA PCA3 and MALAT1, was abruptly used. Many readers would not know what these RNAs are. The authors should describe why they focus on these lncRNAs.  

Response:

Thank for your comment. We introduced these two lncRNAs in detail in the introduction section. For your review, the added information was also attached below. The following sentences were added in the section of Introduction.

“We previously evaluated the diagnostic performance of urinary MALAT1 and PCA3 in Chinese population and demonstrated that both urinary MALAT11 and PCA32 could be a useful biomarker for the early detection of PCa. Due the easy catch, non-invasive source of specimens and stable, detectable source of biomarkers, we proposed that urinary exosomal MALAT1 and PCA3 could be the promising biomarkers for the improvement of PCa detection.”

Materials and Methods -Is the biopsy protocol unified among study institutions? Differences in biopsy counts will lead to differences in cancer positive rate. Subcategory titles, WB, NTA, TEM, and qPCR, should be written with a full version. - Authors performed the qPCR method in this study. Please explain how to perform the reverse transcription. - In paragraph 2.7. qPCR, the authors state, "The PCA3 or MALAT1 score was calculated as 2-Ct(PCA3/MALAT1-Ct(ACTB)*1000." The closing parenthesis corresponding to the first parenthesis is missing. - In paragraph 2.8. Statistics, the authors state statistical methods only for the lncRNA score, the regression model of LncRNA assay, the ROC curve, and the DCA curve. The authors should describe all statistical methods using in this study. (e.g., statistical analyses performed for clinical characteristics of participants.)  

Response:

Thank for your valuable comments. We carefully modified the section of Materials and Methods according to your comments. Please refer to the section of Materials and Methods.

Results -The authors reported that the PCA3 score and the MALAT1 score in the cancer positive group/non-aggressive group were significantly higher than that in the cancer negative group/clinically significant group, but the specific figures are not shown in the text and the figure. Please explain the specific figures. –

Response:

Thank for your comment. The PCA3 score and MALAT1 score were illustrated in the Figure 2a and 2b.

The authors analyze the effectiveness of the PCA3 score and the MALAT1 score by the ROC curve. Please indicate the cut-off values for the PCA3 score and the MALAT1 score, and describe why the authors select that cut-off value as the best optimal. According to Figure 2a and b, the PCA3 score and the MALAT1 score seems to have a vast range.  

Response:

Thank for your valuable comment. Since the main purpose of the study is to establish the urinary exosomal lncRNA assay for the improvement of PCa detection, we only set the cutoff value of the lncRNA assay rather than each cutoff value of PCA3 or MALAT1 score. Here, we take the cutoff values at the sensitivity of 90% and 95%, respectively, according to the previous studies3, 4, to observe the specificity, NPV (Negative Predictive Value) and PPV (Positive Predictive Value) of the assay. PPV and NPV of prostate cancer diagnosis of lncRNA assay is 0.761(95%CI: 0.758-0.764) and 0.74(95%CI: 0.738-0.742). PPV and NPV of clinically significance prostate cancer diagnosis of lncRNA assay is 0.673(95%CI: 0.669-0.677) and 0.784(95%CI: 0.784-0.786).

Discussions -The authors should describe the limitations of this study.

Response:

Thank for your valuable comment. We carefully describe the limitation of this study. Please refer to the section of Discussion.

For your review, the attached the information below.

“However, there still are a few limitations. First, the sample size was limited. A further multicenter, prospective, large-scale study is needed to confirm our findings. Second, we did not perform head-to-head comparisons between the lncRNA assay and other emerging assays, including MiPS, SelectMDx, EPI, etc. Finally, given the remarkable distinction of genetic alteration signatures between Asian and Western populations, additional studies should be launched to compare the clinical value of our lncRNA assay in Asian patients compared to that in Western cohorts.”

  1. Wang, F.; Ren, S.;  Chen, R.;  Lu, J.;  Shi, X.;  Zhu, Y.;  Zhang, W.;  Jing, T.;  Zhang, C.;  Shen, J.;  Xu, C.;  Wang, H.;  Wang, H.;  Wang, Y.;  Liu, B.;  Li, Y.;  Fang, Z.;  Guo, F.;  Qiao, M.;  Wu, C.;  Wei, Q.;  Xu, D.;  Shen, D.;  Lu, X.;  Gao, X.;  Hou, J.; Sun, Y., Development and prospective multicenter evaluation of the long noncoding RNA MALAT-1 as a diagnostic urinary biomarker for prostate cancer. Oncotarget 2014, 5 (22), 11091-102.
  2. Wang, F. B.; Chen, R.;  Ren, S. C.;  Shi, X. L.;  Zhu, Y. S.;  Zhang, W.;  Jing, T. L.;  Zhang, C.;  Gao, X.;  Hou, J. G.;  Xu, C. L.; Sun, Y. H., Prostate cancer antigen 3 moderately improves diagnostic accuracy in Chinese patients undergoing first prostate biopsy. Asian journal of andrology 2017, 19 (2), 238-243.
  3. McKiernan, J.; Donovan, M. J.;  O'Neill, V.;  Bentink, S.;  Noerholm, M.;  Belzer, S.;  Skog, J.;  Kattan, M. W.;  Partin, A.;  Andriole, G.;  Brown, G.;  Wei, J. T.;  Thompson, I. M., Jr.; Carroll, P., A Novel Urine Exosome Gene Expression Assay to Predict High-grade Prostate Cancer at Initial Biopsy. JAMA oncology 2016, 2 (7), 882-9.
  4. McKiernan, J.; Donovan, M. J.;  Margolis, E.;  Partin, A.;  Carter, B.;  Brown, G.;  Torkler, P.;  Noerholm, M.;  Skog, J.;  Shore, N.;  Andriole, G.;  Thompson, I.; Carroll, P., A Prospective Adaptive Utility Trial to Validate Performance of a Novel Urine Exosome Gene Expression Assay to Predict High-grade Prostate Cancer in Patients with Prostate-specific Antigen 2-10ng/ml at Initial Biopsy. European urology 2018, 74 (6), 731-738.
